# Adipocytes Promote Breast Cancer Cell Survival and Migration through Autophagy Activation

**DOI:** 10.3390/cancers13153917

**Published:** 2021-08-03

**Authors:** Dorine Bellanger, Cléa Dziagwa, Cyrille Guimaraes, Michelle Pinault, Jean-François Dumas, Lucie Brisson

**Affiliations:** Inserm UMR1069, Nutrition, Growth and Cancer, University of Tours, 37032 Tours, France; dorine.bellanger@inserm.fr (D.B.); clea.dziagwa@uliege.be (C.D.); cyrille.guimaraes@univ-tours.fr (C.G.); michelle.pinault@univ-tours.fr (M.P.); jean-francois.dumas@univ-tours.fr (J.-F.D.)

**Keywords:** adipose tissue, autophagy, breast cancer, cell communication, fatty acid

## Abstract

**Simple Summary:**

Breast tumours are in direct contact with the adipose tissue of the mammary gland. Although the interactions between breast cancer cells and adipocytes that secrete tumour-promoting factors are well known, the molecular mechanisms remain under investigation. The aim of our study was to understand whether and how adipocytes regulate a cell-recycling pathway in breast cancer cells—autophagy. We show that adipocytes promote autophagy in breast cancer cells through the acidification of lysosomes, leading to cancer cell survival in nutrient-deprived conditions and to cancer cell migration. In this study, we have identified a new mechanism, which can link adipose tissue with breast cancer progression.

**Abstract:**

White adipose tissue interacts closely with breast cancers through the secretion of soluble factors such as cytokines, growth factors or fatty acids. However, the molecular mechanisms of these interactions and their roles in cancer progression remain poorly understood. In this study, we investigated the role of fatty acids in the cooperation between adipocytes and breast cancer cells using a co-culture model. We report that adipocytes increase autophagy in breast cancer cells through the acidification of lysosomes, leading to cancer cell survival in nutrient-deprived conditions and to cancer cell migration. Mechanistically, the disturbance of membrane phospholipid composition with a decrease in arachidonic acid content is responsible for autophagy activation in breast cancer cells induced by adipocytes. Therefore, autophagy might be a central cellular mechanism of white adipose tissue interactions with cancer cells and thus participate in cancer progression.

## 1. Introduction

Breast cancer is the most common cancer in women worldwide. Although vast improvements have been made in diagnosis, treatment is still challenging for metastatic and resistant breast cancers. Cancer progression relies mainly on cell proliferation and adaptation to stressful conditions, and on metastatic dissemination. Indeed, tumours are complex multi-cellular tissues submitted to high selective pressure. In order to survive, cancer cells need to adapt to the surrounding microenvironment composed of physical factors such as hypoxia, nutrient deprivation and of non-cancer cells such as endothelial cells, fibroblasts, immune cells or adipocytes. The metabolic plasticity of cancer cells allows them to use different metabolic substrates (glucose, lactate, amino acids, fatty acids) to produce ATP and NADPH to regulate redox balance and the building blocks needed for a high proliferation rate [1]. Besides the common alteration in glucose metabolism (Warburg effect), fatty acids have been recognised as important fuels for cancer cells [2]. In addition to their metabolic function, fatty acids play important roles as signalling molecules and in membrane structure. Cancer cells take advantage of two main fatty acid sources: *de novo* fatty acid synthesis, which is increased in several cancer cells including liver, prostate, colon and breast cancers [3], and the uptake from the surrounding extracellular microenvironment through transport proteins such as CD36 or Fatty Acid Binding Proteins (FAPBs) [4].

The role of adipose tissue in breast cancer progression is becoming evident due to the recognition over the last few decades that adipose tissue is not only a quiescent tissue for the storage of fatty acids, but has metabolic and endocrine functions with the secretion of lipids or protein factors named adipokines [5,6]. Cooperation between adipocytes and cancer cells has been demonstrated to participate in cancer progression in several tumours, such as ovarian, prostate, colon, liver, skin and breast cancers [7]. In addition, the adipose tissue dysfunction observed in obesity is now recognised as an important risk factor for breast cancer [8]. In all stages of cancer development, breast tumours are in proximity to the white adipose tissue of the mammary gland. On one hand, cancer cells decrease adipocyte lipid content and induce the overexpression of proteases and proinflammatory cytokines, leading to an activated phenotype called cancer-associated adipocytes [9]. On the other hand, adipocytes interact with cancer cells mainly through adipokine secretion, such as IL-6 or Leptin [7], or through the modulation of cancer cell metabolism [10,11]. In breast cancer cells, it has been demonstrated that adipokine and fatty acid secretion by adipocytes increases cancer cell proliferation, migration and invasion [9,11,12]. Interestingly, not only adipose tissue proximity and quantity, but also its composition is associated with breast cancer progression [13]. In particular, a low level of long-chain n-3 polyunsaturated fatty acids in breast adipose tissue is associated with poor prognosis parameters such as tumour multifocality [14]. Therefore, it is important to understand how lipids, depending on their composition, are involved in the crosstalk between adipose tissue and cancer cells and can promote breast cancer progression.

Recently, lipids have been associated with the regulation of autophagy [15], a cell survival process activated by the adverse conditions of the tumour microenvironment and having a key role in tumour progression [16,17]. This process allows the degradation and the recycling of proteins and organelles following the fusion between an isolation vesicle called an autophagosome and a lysosome that provides the hydrolytic enzymes and the acidic pH needed for their activity. In breast cancer, markers of autophagy are associated with tumour aggressiveness and poor prognosis [18,19], suggesting the important role of autophagy in cancer progression. Autophagy is also activated by chemo and radiotherapy anticancer treatments and may be responsible for the acquisition of cancer cell resistance to these treatments [20,21]. In general, lipids, mostly phospholipids, are associated with the regulation of autophagy since they are direct constituents of the autophagosome membrane, and they participate in the autophagosome formation and fusion with lysosomes, mainly through the regulation of the ubiquitin-like protein conjugation system leading to LC3 conjugation to phosphatidylethanolamine [15]. In addition, specific fatty acids, depending on their unsaturation, can regulate autophagy [22,23,24]. It has been demonstrated that free fatty acids derived from adipocytes induce AMPK activation in breast cancer cells [11], which is a major regulator of autophagy. This recent association between lipids and autophagy supports the hypothesis that autophagy could be a central cellular mechanism of white adipose tissue interactions with cancer cells, participating in cancer progression.

In this study, we aimed to understand the role of fatty acids in the cooperation between adipocytes and breast cancer cells, with a focus on autophagy. We used an indirect co-culture of breast cancer cells with adipocytes, using a transwell system, to model in vitro the influence of adipocytes on breast cancer cells.

## 2. Materials and Methods

### 2.1. Cells and Chemicals

OP9 murine embryonic stem cells (CRL-2749), murine mammary cancer cells 4T1 (CRL-2539) and human breast cancer cells MCF7 (HTB-22) and MDA-MB-231 (HTB-26) were obtained from LGC standards (Molsheim, France).

OP9 cells were grown in Alpha-MEM medium (BE02-002, Ozyme, Saint-Quentin-en-Yvelines, France) supplemented with 20% Foetal Bovine Serum (FBS, Ozyme, Saint-Quentin-en-Yvelines, France). The human breast cancer cell lines MCF-7 and MDA-MB-231 were grown in Dulbecco’s modified Eagle’s medium (DMEM, BE12-702, Ozyme, Saint-Quentin-en-Yvelines, France) supplemented with 5% FBS. 4T1 mouse mammary cancer cells were grown in RPMI-1640 (10-013-CV, Corning, Amsterdam, The Netherlands) supplemented with 10% FBS. All cells were incubated in a 37 °C humidified atmosphere containing 5% CO_2_.

Unless stated otherwise, all chemicals were from Sigma-Aldrich (Saint Quentin Fallavier, France). Fluorescent probes (DQ red BSA, LysoSensor) were purchased from Thermofisher (Illkirch, France) and Magic Red cathepsin B kit from Clinisciences (Nanterre, France)

### 2.2. Differentiation of OP9 and Production of Conditioned Medium

OP9 cells were differentiated in adipocytes as previously demonstrated [25]. Confluent OP9 cells were treated with a cocktail containing 0.25 μM dexamethasone, 175 nM insulin, 0.5 mM IBMX in complete Alpha-MEM medium. After 4 days, cocktail was replaced by complete Alpha-MEM containing 175 nM insulin for 3 days. Cells were used in the next 7 days after completion of the differentiation, after removal of insulin and washes in complete Alpha-MEM medium without insulin. Differentiation was controlled microscopically by the presence of large lipid droplets in more than 50% of the cells. OP9 conditioned medium was produced in Alpha-MEM without FBS by incubation during 24 h with differentiated OP9. Conditioned medium was harvested and cellular debris was removed by centrifugation. Alpha-MEM without FBS was used as a control.

### 2.3. Co-Culture System

Cancer cells and adipocytes were co-cultivated using a transwell culture system (1 μm pore size PET transparent membrane, Merck Millipore, Molsheim, France or Falcon Corning, Amsterdam, The Netherlands). Cancer cells were seeded on the bottom of the wells and grown with co-culture inserts containing differentiated (Dif) or non-differentiated (ND) OP9 cells. Co-culture was maintained for 48 h in respective complete medium with and without 20 µM chloroquine for the last 18 h. Arachidonic acid was used at 20 µM during 48 h of co-culture. For nutrient deprivation experiments, cancer cells were seeded in complete medium 24 h before the co-culture experiment. Then, cancer cells and OP9 cells were washed in HBSS to remove all medium and maintained in co-culture for 48 h in HBSS medium.

For fatty acid transfer experiment, OP9 ND and OP9 Dif were loaded with BODIPY FL C_16_ (Thermofisher, Illkirch, France) in complete Alpha-MEM without and with 175 nM insulin, respectively, for 5 h. Then, OP9 cells were washed 4 times with complete Alpha-MEM to completely remove fluorescent fatty acids in solution. MCF7 and 4T1 breast cancer cells were grown in presence of these loaded cells for 48 h. Staining was observed using Nikon TI-S (Champigny-sur-Marne, France) epifluorescence microscope.

### 2.4. Oil Red O Staining

Oil red O was used to stain neutral lipids in differentiated OP9 cells and in cancer cells. Cells grown on coverslips or on µ-Plates (Ibidi Clinisciences, Nanterre, France) were fixed with 4% paraformaldehyde for 15 min. After 3 washes with PBS and 3 washes with distilled water, cells were stained with 0.3% Oil red O solution for 15 min. Oil red O was removed and wells were washed 5 times with water to remove excess solution. Coverslips were mounted with Vectashield Vibrance with DAPI (Eurobio, Les Ulis, France). Oil red O staining was observed using Nikon TI-S (Champigny-sur-Marne, France) epifluorescence microscope and fluorescence intensity of lipid droplets was measured using ImageJ and normalised by cell number on minimum 5 images per condition.

### 2.5. RNA Extraction, RNA Sequencing and RT-qPCR

Total RNA was extracted using classical phenol chloroform method (Trizol Thermofisher, Illkirch, France) according to manufacturer’s protocol. Quality and quantity of RNA extracted were controlled using Take3 BioTek Spectrophotometer. Phenol contamination was cleaned by NucleoSpin Gel and PCR clean up (Macherey Nagel, Hoerdt, France), according to manufacturer’s recommendations. Residual genomic DNA was degraded by rDNase enzyme (Macherey Nagel, Hoerdt, France) for 10 min at 37 °C and inhibited by 2 mM EGTA.

RNA-Sequencing was performed by the Genomic platform of the Institut Cochin. Libraries were prepared using TruSeq mRNA Stranded Library Prep Kit (Illumina, San Diego, CA, USA) and controlled using Agilent 2100 Bioanalyzer (Agilent Technologies, Les Ulis, France). Libraries were sequenced on Illumina NextSeq 500 (Illumina, San Diego, CA, USA) using 75 base-length. Star algorithm was used to align reads against the reference genome (Ensembl GRCh38 release 96). Statistical analysis was performed using R and RSEM and DESeq2 packages. Genes differentially expressed between co-culture with OP9 Dif vs. OP9 ND and OP9 ND vs. without co-culture were represented in Venn diagram using Gdata and VennDiagram packages for R.

Reverse transcription was performed using the PrimeScript RT Reagent Kit (Ozyme, Saint-Quentin-en-Yvelines, France). Quantitative PCR was performed using SYBR qPCR Premix Ex Taq (Ozyme, Saint-Quentin-en-Yvelines, France) and CFX CONNECT (Bio-rad, Marnes-La-Coquette, France). After primer efficiency testing, 50 ng of cDNA was used for qPCR experiments. HPRT and β-actin were used as housekeeper genes after selection using Normfinder. Data were analysed with CFX maestro software (Bio-rad, Marnes-La-Coquette, France) and results were normalised by OP9 ND condition, using ΔΔCt relative quantification method.

### 2.6. Western Blot

Cells were lysed in lysis buffer containing 50 mM Tris pH 7.4, 150 mM NaCl, 1% Triton X-100, 0.05% sodium deoxycholate, 1 mM EDTA, 0.1% SDS and protease inhibitor cocktail (Sigma-Aldrich, Saint Quentin Fallavier, France). After centrifugation (10,000× *g* for 10 min at 4 °C) to remove cellular debris, supernatant was collected. Colorimetric protein assays were performed with Pierce BCA protein assay kit (Thermofisher, Illkirch, France) according to manufacturer’s protocol. Proteins, diluted in Laemmli buffer, were loaded on Mini-Protean TGX 4-20% (Bio-rad, Marnes-La-Coquette, France) and transferred on PVDF membrane using Trans-Blot Turbo RTA Transfer Kit and Trans-Blot Turbo Transfer System (Bio-rad, Marnes-La-Coquette, France). Membranes were saturated for 1 h at room temperature with 5% nonfat dry milk. Primary antibodies were LC3/MAP1LC3B antibody (1/2000, NB600-1384, Novusbio Bio-Techne, Lille, France), p62/SQSTM1 (1/1000, H00008878-M01, Novusbio Bio-Techne, Lille, France), β-actin (1/3000, Clone AC-15, Sigma-Aldrich, Saint Quentin Fallavier, France) and Beclin1 (1/1000, 3738S, Ozyme, Saint-Quentin-en-Yvelines, France). Secondary HRP-conjugated antibodies were goat anti-mouse (1/5000, Santa Cruz Tebu-Bio, Le Perray-en-Yvelines, France) and goat anti-rabbit (1/10000, Jackson Immunoresearch Interchim, Montluçon, France). Chemiluminescence was detected with the SuperSignal West Pico Plus kit (Thermofisher, Illkirch, France) and ChemiDoc Imager (Bio-rad, Marnes-La-Coquette, France). The protein expression of LC3 and p62 was analysed using ImageJ and normalised to control condition. Autophagic flux was calculated as the difference in LC3 expression between conditions with and without chloroquine.

### 2.7. siRNA and Plasmid Transfections

Cells were transfected with siRNA against Beclin1 (siBCN1 CCCGUGGAAUGGAAUGAGATT) using lipofectamine RNAiMAX (Thermofisher, Illkirch, France) according to the reverse manufacturer protocol 24 h before the co-culture experiment.

LC3-mRFP-GFP plasmid (Plasmid 21074: ptfLC3) [26] was transfected 24 h before the end of the co-culture experiment with TransiT-2020 transfection reagent (Mirus Euromedex, Souffelweyersheim, France). After fixation with PFA 4% for 15 min, cells were observed under a Nikon TI-S epifluorescence microscope (Champigny-sur-Marne, France). Yellow and red vesicles were counted using the Red and Green Puncta Colocalisation Macro for ImageJ (developed by Daniel J. Shiwarski, Ruben K. Dagda and Charleen T. Chu) to identify autophagosome and autolysosome number on minimum 5 images and 15 cells per condition.

### 2.8. Intracellular Degradation

Intracellular degradation was measured using DQ-BSA for MCF7 and Magic Red cathepsin B kit for 4T1 cells. DQ-BSA (25 µg/mL) and Magic Red (1/2000) were added to culture medium 1 h before the end of the co-culture experiment. Cells were immediately observed with a Nikon TI-S (Champigny-sur-Marne, France) epifluorescence microscope. Fluorescence intensity was measured using ImageJ and normalised by cell number observed using bright field on minimum 5 images per condition.

### 2.9. Lysosome pH and LAMP1 Staining

LysoSensor^TM^ Green DND-189 (Thermofisher, Illkirch, France) was used to measure lysosomal pH. Cells were incubated with 1 µM LysoSensor in HBSS for 30 min at 37 °C then washed with HBSS. Epifluorescence microscopy was performed with a Nikon TI-S (Champigny-sur-Marne, France) and fluorescence intensity was measured using ImageJ and normalised to cell number observed using bright field on minimum 5 images per condition. Bafilomycin was used as a control of lysosomal alkalinisation.

Standard immunohistochemistry protocol was followed for LAMP1 staining. Cells were fixed with PFA 4% for 15 min, then washed in PBS and permeabilised with 0.3% Triton for 5 min. After blocking with 5% BSA, primary antibody against LAMP1 (1/20, ab25630, Abcam, Paris, France) was incubated for 1 h. Secondary antibody was AlexaFluor488 or AlexaFluor568 anti-mouse (1/1000, Thermofisher, Illkirch, France). Preparations were mounted using Vectashield Vibrance with DAPI (Eurobio, Les Ulis, France). Epifluorescence microscopy was performed with a Nikon TI-S (Champigny-sur-Marne, France).

### 2.10. Oxygen Consumption Measurement

Cellular oxygen consumption was measured using high resolution respirometry Oroboros O2K, as described previously [27]. Four million intact cells (without co-culture or with co-culture with OP9 ND or OP9 Dif for 48 h) were loaded in 2 mL of respective medium without FBS in the Oroboros chamber. Basal oxygen consumption was measured, then 100 µM etomoxir was added to the chamber and after stabilisation 2 µM of antimycin A. The total mitochondrial oxygen consumption was obtained by subtracting the oxygen consumption after antimycin A to the basal respiration. The fatty acid-related mitochondrial oxygen consumption was obtained by subtracting the oxygen consumption after etomoxir to the basal respiration.

### 2.11. Lipid Analysis Using High Performance Thin Layer Chromatography (HPTLC) and Gas Chromatography (GC)

Fifteen million 4T1 and MCF7 cells were collected, washed, and pelleted. Lipid extraction was performed according to the method of Bligh and Dyer [28].

HPTLC was performed as previously described [29,30]. Briefly, lipid extracts were spotted on HPTLC precoated silica gel glass plates 60F254 (20 × 10 cm; 1.05642.0001; Merck Millipore, Molsheim, France) using a Camag autosampler ATS4 sample applicator (Camag, Muttenz, Switzerland) and compared with standard calibration curves. Mobile phases were hexane/diethyl ether/acetic acid (70/30/1, *v*/*v*) for free fatty acids and chloroform/ethanol/triethylamine/water (3/3.5/3.5/0.7, *v*/*v*) for phospholipids. After migration, phospholipid plates were immersed for 2 min in staining solution (0.5% copper sulphate in 1.11 M orthophosphoric acid) and free fatty acid plates were immersed for 1 min in staining solution containing 50% perchloric acid. Plates were dried for 2 h at room temperature. Lipid staining by carbonisation was obtained by heating the phospholipid plates for 15 min at 155 °C and the free fatty acid plates for 16 min at 160 °C on the TLC plateHeater. Each batch of silica glass plates has been validated to optimise lipid staining and carbonisation. Densitometric analysis was performed using a TLC/HPTLC video-densitometer (TLC Visualizer 2 Camag, with Videoscan software, Camag, Muttenz, Switzerland). Phospholipid and free fatty acid spot intensity was expressed according to their surface in arbitrary units.

For GC, total lipids were separated using preparative silica gel thin layer chromatography (one-dimensional TLC) plates (LK5, 20 × 20 cm, Merck Millipore, Molsheim, France) and hexane/diethyl ether/acetic acid (70/30/1, *v*/*v*) as a solvent [31,32]. Phospholipid spots were scraped and collected in screw-cap glass tubes. Fatty acids from phospholipids were prepared as fatty acid methyl esters before gas chromatography analysis (GC-2010plus, Shimadzu Scientific instruments, Marne-la-Vallée, France) using a BPX70 capillary column (0.25 µm, 60 m × 0.25 mm ID, SGE, Chromoptic SAS, Courtabœuf, France). Hydrogen was used as carrier gas with a constant pressure (220 kPa). After an on-column injection of sample at 60 °C, oven temperature increased from 60 to 220 °C. Fatty acids were detected by a Flame Ionisation Detector at 225 °C and identified by comparison of their retention times with commercial standards (Supelco 37 Fatty Acid Methyl Ester mix, Sigma-Aldrich, Saint Quentin Fallavier, France). Fatty acid ester levels were expressed as the percentage of total integrated peak area using the GC solutions software (Shimadzu Scientific Instruments, Marne-la-Vallée, France).

### 2.12. Viability Assays

Cell viability, survival in HBSS and short-term toxicity were evaluated using standard sulforhodamine B (SRB) method after 48 h and 24 h treatments, respectively. Briefly, cells were fixed with 50% trichloroacetic acid for 1 h at 4 °C and stained for 15 min with 0.4% SRB solution. Cells were washed 3 times with 1% acetic acid and dye was dissolved with 10 mM trisbase solution over 10 min. Absorbance at 540 nm was read using BioTek Spectrophotometer.

### 2.13. Migration and Invasion Assays

Cancer cell migration was measured using standard transwell assay using 8 µm pore polyethylene terephthalate membrane inserts (Corning, Amsterdam, The Netherlands) and invasion using the same inserts covered with Matrigel matrix (Corning, Amsterdam, The Netherlands). A total of 40,000 4T1, 60,000 MCF7 and 20,000 MDA-MB-231 for migration or 40,000 for invasion cells, grown for 48 h in co-culture with OP9 ND and OP9 Dif, were seeded on the top of the insert with and without 20 µM of chloroquine in respective complete medium. The bottom of the insert was filled with medium containing 20% of SVF for 4T1 or 10% of SVF for MCF7 and MDA-MB-231 with and without 20 µM of chloroquine. Migrating cells at the lower surface of the insert were stained with DAPI and 5 representative pictures per insert were taken with a Nikon TI-S microscope (Nikon, Champigny-sur-Marne, France). The total number of nuclei per image was counted after thresholding using the particle analysis tool of ImageJ.

### 2.14. Statistical Analysis

Statistical analyses were performed using GraphPad Prism V6 software. Normality was tested using D’Agostino and Pearson omnibus test. When the normality test was passed, parametric tests were used: one sample *t* test or one-way ANOVA followed by Holm–Sidak’s multiple comparisons test, and data were expressed as mean ± SEM. When normality test did not pass or was not possible for small sample experiments, nonparametric tests were used: Mann–Whitney test, Wilcoxon Signed Rank test or Kruskal–Wallis followed by Dunn’s Multiple Comparison test, and data were expressed as median ± interquartile range. Two-way ANOVA followed by Bonferroni’s multiple comparisons test was used for multiple comparisons of fatty acid composition. Statistical tests and data representation are indicated in the figure legends. Statistical significance is indicated as: * *p* < 0.05; ** *p* < 0.01 and *** *p* < 0.001. NS stands for not statistically different.

## 3. Results

### 3.1. Co-Culture Model for Adipocyte and Breast Cancer Cell Interactions

We aimed to understand the interactions between adipocytes and breast cancer cells. Therefore, we characterised a co-culture model between adipocytes obtained after the differentiation of OP9 murine stem cells and breast cancer cells MCF7, 4T1 or MDA-MB-231. We used indirect co-culture with a transwell system, which allows cell communications without direct contact between the two cell types. The co-culture of breast cancer cells with differentiated OP9 (OP9 Dif) was compared with the co-culture with non-differentiated OP9 cells (OP9 ND). As shown in Appendix A, the differentiation of OP9 cells leads to cells filled with large lipid droplets, and to an increase in adipocyte markers Perilipin 1, FABP4 and CD36. The co-culture of breast cancer cells with OP9 Dif induces a decrease in lipid content in adipocytes (Appendix A) and an accumulation of lipid droplets in breast cancer cells compared to the co-culture with non-differentiated OP9 cells (Figure 1A,B and Appendix A). These results suggest a transfer of fatty acids from OP9 Dif to breast cancer cells, as reported with co-culture models using 3T3-F442A adipocytes and breast cancer cells [11]. In order to confirm this fatty acid transfer between the two cell types, we loaded fluorescent fatty acid (BODIPY FL C_16_) in OP9 Dif (Appendix A) and measured the fluorescence in breast cancer cells after co-culture. We observed a high amount of fluorescent fatty acids in breast cancer cells after the co-culture with fluorescent fatty acid-loaded OP9 Dif compared with OP9 ND (Figure 1C). The staining is diffuse with punctuated structures mainly in MCF7, suggesting that transferred fatty acids are located both in cellular membranes and in lipid droplets. Altogether, these results show that this co-culture is a proper model to investigate the cooperation between adipocytes and cancer cells, and in particular the role of fatty acids transferred from adipocytes to breast cancer cells.

### 3.2. Adipocytes Activate Autophagy in Breast Cancer Cells

Since fatty acids have been linked with autophagy [15], a cell survival pathway involved in breast cancer progression, we used this co-culture model to investigate the influence of adipocytes on autophagy in breast cancer cells. As shown in Figure 2A,B and in Appendix A, OP9 Dif increases autophagic flux, measured by the turnover of LC3II, a key autophagic protein, with and without the inhibition of autophagy by chloroquine in MCF7, 4T1 and MDA-MB-231 breast cancer cells compared to the co-culture with OP9 ND. This effect is associated with a reduction in the autophagic cargo receptor p62 in breast cancer cells in co-culture with OP9 Dif compared to OP9 ND (Figure 2A,B and Appendix A). Co-culture with OP9 ND has no effect on autophagic flux and p62 expression compared to cancer cells alone (Appendix A).

Then, we wanted to understand which step of the autophagic process was promoted by adipocytes. First, we investigated the crucial step of autophagosome fusion with lysosomes using transfection with the LC3-GFP-mRFP plasmid. The number of autophago-lysosomes (only red dots) increases in MCF7 and 4T1 breast cancer cells grown with OP9 Dif, whereas there is no difference in the number of autophagosomes (yellow dots) (Figure 3A,B). The treatment of co-cultures with autophagy inhibitor chloroquine prevents the increase in autolysosome number in 4T1 and MCF7 induced by OP9 Dif (Figure 3A,B). The last step of autophagy, which depends on autophagosome fusion with lysosomes and on lysosomal enzymes, is intracellular degradation. Intracellular degradation was measured in 4T1 and MCF7 cells using specific fluorescent substrates, respectively, cysteine cathepsin B substrate and general protease substrate. We observe an increase in the fluorescence of the intracellular degradation substrates in 4T1 and MCF7 breast cancer cells grown with OP9 Dif compared to OP9 ND (Figure 3C). Finally, the acidic lysosomal pH is a key element for intracellular degradation and autophagic activity. Lysosomal pH was measured using a LysoSensor probe. Bafilomycin, as a positive control, decreases the LysoSensor signal because of lysosomal alkalinisation (Figure 3D). In conditions of co-culture with OP9 Dif, the lysosomal pH is more acidic in MCF7 and 4T1 breast cancer cells compared to the co-culture with OP9 ND (Figure 3D). Co-culture with OP9 ND has no effect on autophagosome fusion with lysosomes, on intracellular degradation and on lysosomal pH compared to cancer cells alone (Appendix A and Figure 3D). Overall, these results show that adipocytes increase autophagy in breast cancer cells through an increase in autophagosome fusion with lysosomes and through the acidification of lysosomal pH.

### 3.3. Role of Fatty Acids in the Promotion of Autophagy by Adipocytes

In order to understand how autophagy is regulated in breast cancer cells by adipocyte proximity, we first checked the metabolic role of fatty acids transferred from adipocytes to breast cancer cells. Indeed, autophagy is finely regulated by the availability of metabolic substrates and the ATP/ADP ratio. In breast cancer cells, it was previously demonstrated that free fatty acids transferred from adipocytes induce a metabolic switch toward mitochondrial fatty acid oxidation uncoupled to ATP production [11]. In this co-culture model, the total and fatty acid-dependent mitochondrial oxygen consumptions were not significantly modified in MCF7 and 4T1 cells in proximity to OP9 Dif (Appendix A), ruling out a major role of fatty acids as metabolic fuels to activate autophagy.

Adipokines are known to induce signalling pathways in cancer cells through transcriptional regulations [7]. In addition, free fatty acids are endogenous ligands of membrane or intracellular receptors such as G-protein coupled receptors (FFA1-4) and the peroxisome proliferator-activated receptors (PPARs) transcription factors. The fixation of free fatty acids on these receptors in cancer cells could induce signalling pathways or the transcription of genes involved in autophagy. To determine if transcriptional regulation may be involved, we performed non-targeted RNA sequencing in co-culture conditions in MCF7 cells. We identified 4184 genes differentially expressed between the two control conditions: without co-culture and with co-culture with OP9 ND (Appendix A). Only 20 genes are specifically differentially expressed in MCF7 cells in co-culture with OP9 Dif compared to OP9 ND (Appendix A). Among them, we selected five genes, with the highest regulation and with a potential link with autophagy, to analyse in the other cancer cell line, 4T1. The regulations induced by adipocytes are not conserved in 4T1 cells (Appendix A), ruling out the involvement of the transcriptional regulation of autophagy by adipokines or fatty acids.

Some lipids are linked directly with autophagic membranes, with the fusion of autophagosomes with lysosomes or with the regulation of the autophagy pathway [15,33]. Therefore, we measured lipid species quantity and fatty acid composition in MCF7 and 4T1 breast cancer cells treated with conditioned media from OP9 Dif. First, we do not observe an increase in free fatty acid content in breast cancer cells grown with OP9 Dif conditioned media (Appendix A), suggesting that transferred fatty acids are stored in lipid droplets or integrated through intracellular membranes. During autophagosome formation, phosphatidylethanolamine (PE) is conjugated with LC3 to form LC3II or LC3-PE [34]. Therefore, a modification of PE content in membrane phospholipids could influence autophagy. All classes of phospholipids are increased in MC7 and 4T1 grown with OP9 Dif conditioned media compared to control conditions (Appendix A), but the proportion of phospholipid classes is not significantly modified (Appendix A). This suggests that phospholipid class composition in membranes remains similar in the presence or absence of the OP9 Dif conditioned media. However, the increase in PE content could in part participate in the increase in autophagy induced by adipocytes.

Fatty acids, by integrating into intracellular membranes, could modulate the acidification of lysosomes, which is required for the degradation activity of autophagy. Fatty acid composition in cellular membrane phospholipids was studied in MCF7 and 4T1 cells treated with conditioned media from OP9 Dif (Table 1). Among the fatty acids commonly regulated in 4T1 and MCF7 cells treated with OP9 Dif conditioned media, we observe a decrease in arachidonic acid (AA, 20:4n-6), in docosapentaenoic acid (22:5n-3) and in docosahexaenoic acid (22:6n-3) content in membrane phospholipids compared to control conditions (Table 1). This change in fatty acid composition in membrane phospholipids could participate in the regulation of autophagy in breast cancer cells.

### 3.4. AA Prevents Autophagy Activation by Adipocytes

Among the fatty acids identified in Table 1, a high content of docosahexaenoic acid has been associated with autophagy activation [22], while we observed an activation of autophagy with a low docosahexaenoic acid concentration in phospholipids. Interestingly, arachidonic acid has been linked with lysosomal membrane permeabilisation [35] and consequently with the regulation of lysosomal pH and function. Therefore, we supplemented cells in co-culture with a non-toxic AA concentration (20 µM 48 h) (Appendix A) to assess whether it could be involved in the activation of autophagy by adipocytes. Interestingly, AA supplementation prevents the increase in autophagic flux in MCF7 and 4T1 cells grown in co-culture with OP9 Dif (Figure 4A,B). p62 protein level is not restored by AA supplementation, indicating that other mechanisms might be involved (Figure 4A,B). AA supplementation has no effect on autophagic flux on MCF7 cells grown in co-culture with OP9 ND and induces a small decrease in autophagic flux in 4T1 cells grown in co-culture with OP9 ND (Figure 4A,B).

Since AA has been associated with lysosomal membrane permeabilisation [35], its decrease can participate in lysosome acidification and therefore in autophagy promotion by adipocytes. Consequently, AA supplementation prevents lysosomal acidification induced by adipocytes in MCF7 and 4T1 breast cancer cells (Figure 5A,B). Adipocytes and AA supplementation have no effect on the lysosome number (Appendix A), ruling out an effect of adipocytes and AA on lysosome biogenesis. Finally, the increase in intracellular degradation induced by OP9 Dif in MCF7 and 4T1 breast cancer cells is also prevented by AA supplementation (Figure 5C,D). AA addition in cells in co-culture with OP9 ND does not change lysosomal pH and intracellular degradation (Figure 5A–D). Overall, these results demonstrate that the decrease in AA content in membrane phospholipids participates in autophagy promotion by adipocytes since AA supplementation can prevent this effect.

### 3.5. Autophagy Induction by Adipocytes Promotes Cancer Cell Survival and Migration

In cancer cells, autophagy is mainly associated with cell survival in nutrient deprivation conditions and with cancer cell migration [21]. First, we wanted to understand whether the increase in autophagy in breast cancer cells grown with OP9 Dif promotes cancer cell growth and survival in nutrient deprivation conditions. The effect of OP9 Dif was assessed on cancer cell basal viability in complete medium and survival in nutrient deprivation conditions. OP9 Dif has no significant effect on basal viability, whereas it increases survival in nutrient-deprived conditions (Figure 6A,B and Appendix A). The increase in cell survival, in nutrient deprivation conditions, is prevented by a non-toxic concentration of FABP4 inhibitors (Figure 6C and Appendix A). In addition, no significant effect is observed with FABP4 inhibitors in control and in co-culture conditions (Figure 6C). These results suggest that fatty acids transferred from adipocytes to cancer cells are involved in the promotion of cell survival under nutrient deprivation. Interestingly, fatty acid transport might be a general mechanism for cell survival since FABP4 inhibitors decrease cell survival in control cells (Figure 6C). In addition, the inhibition of autophagy in cancer cells using siRNA against Beclin 1 (siBCN1) decreases cell survival induced by OP9 Dif (Figure 6D and Appendix A), showing that autophagy promotion by adipocytes participates in the increase in cell survival. Similar to the effect of OP9 Dif on autophagy, cell survival is also prevented by AA supplementation of cancer cells co-cultured with OP9 Dif (Figure 6E). AA supplementation also decreases cell survival in 4T1 control cells (Figure 6E), whereas it has no effect on basal cell viability (Appendix A).

Finally, we measured the effect of adipocytes on highly aggressive 4T1 and MDA-MB-231 and less aggressive MCF7 cancer cell migration and invasiveness. In all cell lines, migration and invasion are increased in cells pre-cultured with OP9 Dif compared to OP9 ND (Figure 7A,B and Appendix A). Interestingly, in MCF7 and MDA-MB-231 cells, the increase in migration and invasion induced by OP9 Dif is prevented by autophagy inhibition by chloroquine (Figure 7B and Appendix A). In 4T1 cells, in contrast, only migration but not invasion is prevented by chloroquine treatment (Figure 7A), suggesting that the promotion of cancer cell invasion by adipocytes is partially due to autophagy induction in these cells. In addition, the migration of MCF7 and the invasion of MDA-MB-231 cells are increased by chloroquine treatment, indicating that complex mechanisms linking autophagy and migration/invasion occur in these cell lines.

## 4. Discussion

Cellular cooperation is well described in breast cancer. In particular, adipocytes are known to interact with breast cancer cells through the secretion of adipokines and through the release of fatty acids [7]. Although the role of adipokines in cancer cell proliferation and migration is well characterised, the molecular mechanisms of fatty acids in the cooperation between adipocytes and breast cancer cells remain poorly documented. Only a metabolic role of fatty acids from adipocytes has been demonstrated in breast cancer cells [11,36]. In these studies, fatty acids transferred from adipocytes induce metabolic reprogramming in breast cancer cells with increased mitochondrial fatty acid oxidation uncoupled to ATP production and increased anaerobic glycolysis [11,36]. However, fatty acids are important players in signalling pathways or in membrane structure. In this study, we aimed to understand how fatty acids are involved in the cooperation between breast cancer cells and adipocytes with a particular focus on autophagy.

We first characterised a model of co-culture between breast cancer cells with adipocytes obtained after the differentiation of OP9 murine stem cells. Similar to other models with 3T3 differentiated adipocytes [9], breast cancer cells induce delipidation in OP9 Dif, which is typical of cancer-associated adipocytes. In addition, we have shown that fatty acids are transferred from adipocytes to breast cancer cells, then stored in lipid droplets and also are part of membrane phospholipids.

Using this co-culture model, we have shown that adipocytes increase autophagic flux, autophagosome maturation, lysosomal acidification and intracellular degradation in breast cancer cells (Figure 8). It was previously demonstrated in colon cancer cells that adipocytes or oleic acid treatment increase fatty acid oxidation to stimulate autophagy [37]. Nevertheless, in breast cancer cells, the metabolic use of fatty acids or the induction of transcriptional pathways was not involved in the promotion of autophagy by adipocytes. We report for the first time that adipocytes induce a reorganisation of membrane phospholipid composition, with, in particular, a decrease in arachidonic acid content. The decrease in arachidonic acid content was responsible for autophagy activation in breast cancer cells by adipocytes. Specific fatty acids, depending on their unsaturation, were already linked with autophagy regulation, but results remain controversial for the same fatty acids in different models [22,23,24]. Interestingly, it has been demonstrated that unsaturated fatty acids induce a non-canonical form of autophagy, independent of Beclin1 but requiring the Golgi complex [33]. In our model, the decrease in arachidonic acid n-6 unsaturated fatty acid is associated with an increase in autophagy in breast cancer cells in co-culture with adipocytes. These results are important for the understanding of the cooperation between adipose tissue and breast tumours and might lead to the proposition of nutritional intervention to modulate fatty acid composition in the diet, in order to disrupt the crosstalk between adipocytes and breast cancer cells. However, nutritional intervention could have pleiotropic biological effects through the modification of the balance between n-6 and n-3 polyunsaturated fatty acid or the release of active metabolites [38]. Additionally, in obese patients, this signalling pathway could have important involvement for the promotion of breast tumours.

The role of autophagy in tumours is complex and it is now well known that autophagy has a dual role in cancer progression [17,39]. Autophagy is protective for cancer initiation but in tumours its activation by extracellular stress conditions (hypoxia, low nutrients and growth factors, ROS, and lactate) provides metabolic substrates, which is favourable for cell survival and tumour growth [40,41]. Besides these roles, autophagy has been associated with cancer progression, which depends mainly on metastasis dissemination and resistance to anticancer therapies. During metastasis formation, cancer cells need to acquire several properties to escape from the primary tumour, including migration and degradation of extracellular matrices, and to survive in circulation and in distant sites (anoïkis resistance). Additionally, autophagy has been linked with several of these properties [42]. In breast cancer, markers of autophagy are increased in metastases compared to primary tumours [18] and are associated with poor prognosis [19]. In this breast cancer model, we demonstrated that autophagy stimulation by adipocytes promotes cell survival in nutrient deprivation conditions and increases cell migration, suggesting an important role in cancer progression and metastatic dissemination. Breast cancer cell invasion is also promoted by adipocytes, but autophagy induction seems to not be involved in all cell lines tested. Besides breast cancer, in which adipose tissue proximity occurs since the first steps of cancer initiation, adipose tissue is known to interact with several tumours at latter stages [7]. Therefore, this signalling pathway may play an important role, not only in breast cancer progression but also in the progression of all tumours in contact with adipose tissue, and in cancer cells requiring autophagy for survival.

The molecular mechanism involved in fatty acid transfer into cancer cells is not completely understood. It has been demonstrated that exosomes from adipocytes can transport fatty acids to breast cancer cells [43]. Fatty acids are also carried by several transporters, such as CD36, FABPs and Fatty Acid Transport proteins (FATs). In breast cancer cells, CD36 expression increases with adipocyte conditioned medium and its activity is involved in adipocyte-induced migration and invasion [44]. Interestingly, in an ovarian cancer model, it has been demonstrated that CD36 expression is increased by omental adipocyte proximity and its expression in metastasis-initiating cells is associated with metastasis dissemination and poor prognosis [45,46]. An increased circulating level of FABP4 has been reported in obese breast cancer patients and associated with breast cancer stemness and aggressiveness [47]. FABP5 is also overexpressed in breast cancer cells in contact with adipocytes and correlates with tumour aggressiveness [48]. In this study, we demonstrated that FABP4 activity is required for breast cancer cell survival in nutrient deprivation conditions, both in control conditions and in co-culture with adipocytes. In addition, FABP4 inhibition prevents cell survival induced by adipocytes. Therefore, fatty acid transporters represent an interesting target to limit the promotion of autophagy by adipocytes and reduce cancer cell survival and migration.

## 5. Conclusions

In conclusion, we demonstrated that adipocytes increase autophagy in breast cancer cells to promote cell survival and cell migration. These results suggest that this signalling pathway provides an advantage to breast cancer cells in proximity to adipose tissue, and could be targeted by the inhibition of fatty acid transfer between adipocytes and breast cancer cells.

## Figures and Tables

**Figure 1 cancers-13-03917-f001:**
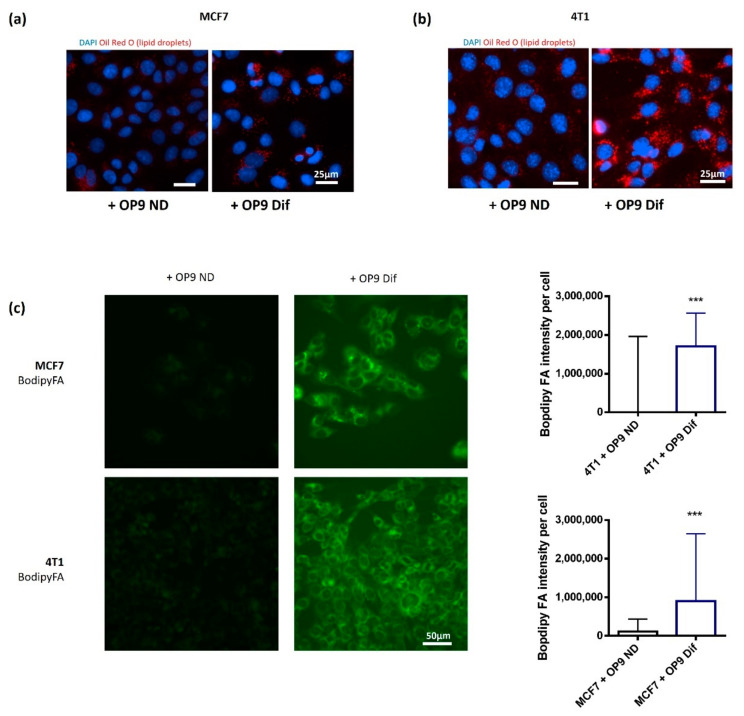
Co-culture with adipocytes increases lipid droplets in breast cancer cells. (**a**,**b**) Lipid droplets were stained using Oil Red O in MCF7 (**a**) and 4T1 (**a**) cells in co-culture with OP9 ND and OP9 Dif for 48 h. DAPI was used to normalise by cell number (representative of *n* = 6). (**c**) BODIPY FL C_16_ in MCF7 and 4T1 breast cancer cells after 48 h of co-culture with BODIPY FL C_16_ loaded OP9 ND and OP9 Dif (Median ± interquartile range, Mann–Whitney test *** *p* < 0.001, *n* = 5).

**Figure 2 cancers-13-03917-f002:**
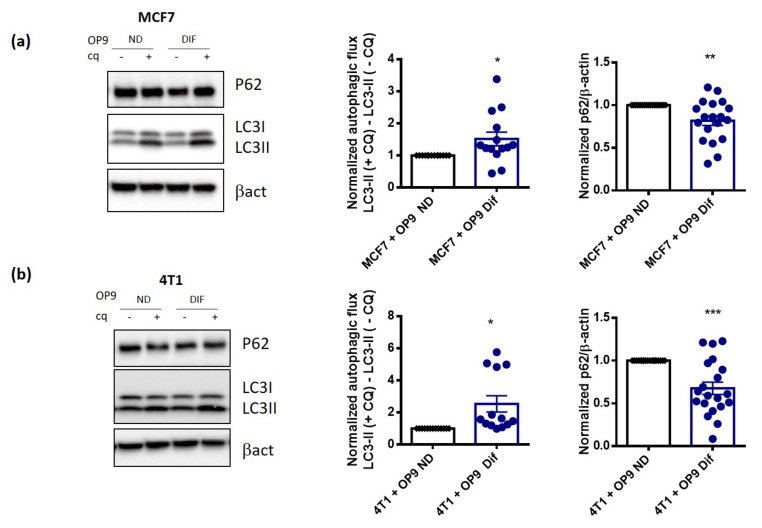
Co-culture with adipocytes activates autophagy in breast cancer cells. (**a**,**b**) Representative western blots and quantification of normalised autophagic flux and p62 protein expression in MCF7 (**a**) and 4T1 (**b**) cells in co-culture with OP9 ND and OP9 Dif for 48 h. To calculate the autophagic flux, 20 µM chloroquine (CQ) was used for 18 h (Mean ± SEM, One sample *t* test, *n* = 13–14 for autophagic flux * *p* < 0.05; ** *p* < 0.01 and *** *p* < 0.001, *n* = 19–20 for p62).

**Figure 3 cancers-13-03917-f003:**
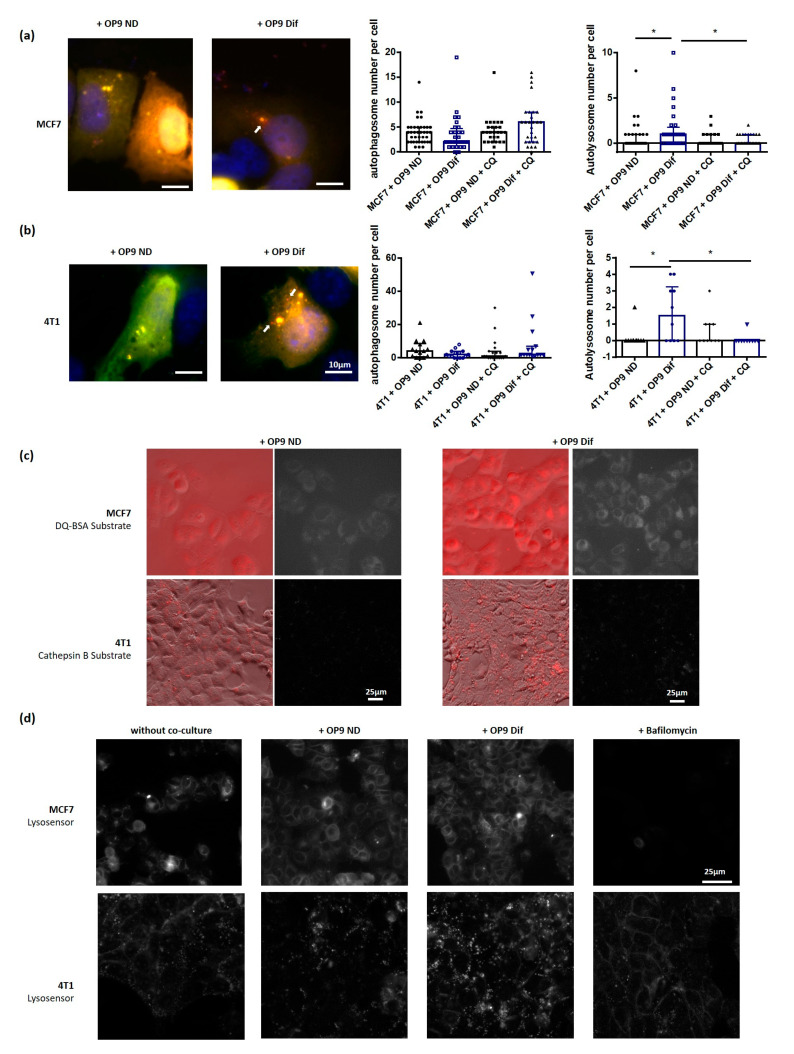
Adipocytes promote autolysosome maturation, intracellular degradation and lysosomal acidification in breast cancer cells. (**a**,**b**) Autophagosome and autolysosome number was measured using the LC3-GFP-mRFP plasmid in MCF7 (**a**) and 4T1 (**b**) cells in co-culture with OP9 ND and OP9 Dif for 48 h. DAPI was used to normalise by cell number. White arrows show red autolysosomes: due to acidic pH, GFP signal is quenched in autolysosomes. In the case of autophagosomes, dots are yellow (GFP and RFP signals). (Median ± interquartile range, Kruskal–Wallis test followed by Dunn’s Multiple Comparison Test * *p* < 0.05, *n* = 3). (**c**) Intracellular degradation, measured in MCF7 cells using DQ-BSA substrate, and in 4T1 cells using Cathepsin B substrate, is increased in proximity to OP9 Dif compared to OP9 ND (*n* = 3). (**d**) LysoSensor probe was used to measure lysosomal pH in MCF7 and 4T1 cells grown in proximity to OP9 ND and OP9 Dif for 48 h and without co-culture. Bafilomycin A1 (100 nM) was used as a positive control for lysosomal pH increase (*n* = 3).

**Figure 4 cancers-13-03917-f004:**
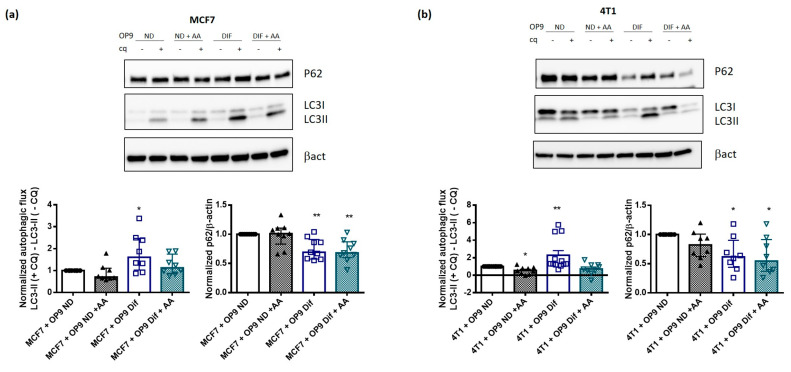
Arachidonic acid supplementation prevents the activation of autophagy by adipocytes in breast cancer cells. (**a**,**b**) Representative western blots and quantification of normalised autophagic flux and p62 protein expression in MCF7 (**a**) and 4T1 (**b**) cells in co-culture with OP9 ND and OP9 Dif, with and without arachidonic acid (AA, 20 µM) for 48 h. To calculate the autophagic flux, 20 µM chloroquine (CQ) was used for 18 h (Median ± interquartile range, Wilcoxon Signed Rank Test * *p* < 0.05 and ** *p* < 0.01, *n* = 8–12).

**Figure 5 cancers-13-03917-f005:**
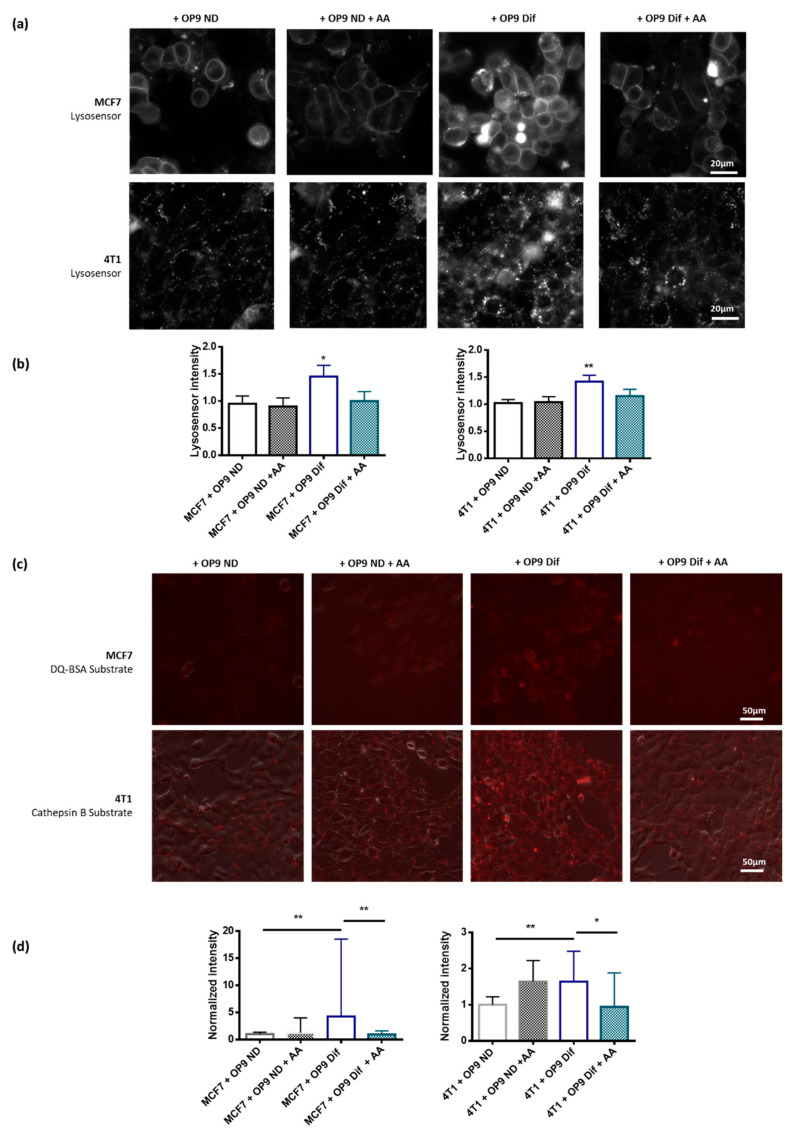
Arachidonic acid supplementation prevents intracellular degradation and lysosomal acidification induced by adipocytes in breast cancer cells. (**a**) Representative images of lysosomal pH measurement using LysoSensor probe in MCF7 and 4T1 cells in co-culture with OP9 ND and OP9 Dif and treated with and without arachidonic acid (AA) 20 µM for 48 h. (**b**) Quantification of A (Mean ± SEM, One sample *t* test * *p* < 0.05 and ** *p* < 0.01, *n* = 5–7). (**c**) Representative images of intracellular degradation measurement using DQ-BSA substrate in MCF7 and Cathepsin B substrate in 4T1 cells in co-culture with OP9 ND and OP9 Dif, treated with and without arachidonic acid (AA) 20 µM for 48 h. (**d**) Quantification of C (Median ± interquartile range, Kruskal–Wallis followed by Dunn’s Multiple Comparison Test * *p* < 0.05 and ** *p* < 0.01, *n* = 6–8).

**Figure 6 cancers-13-03917-f006:**
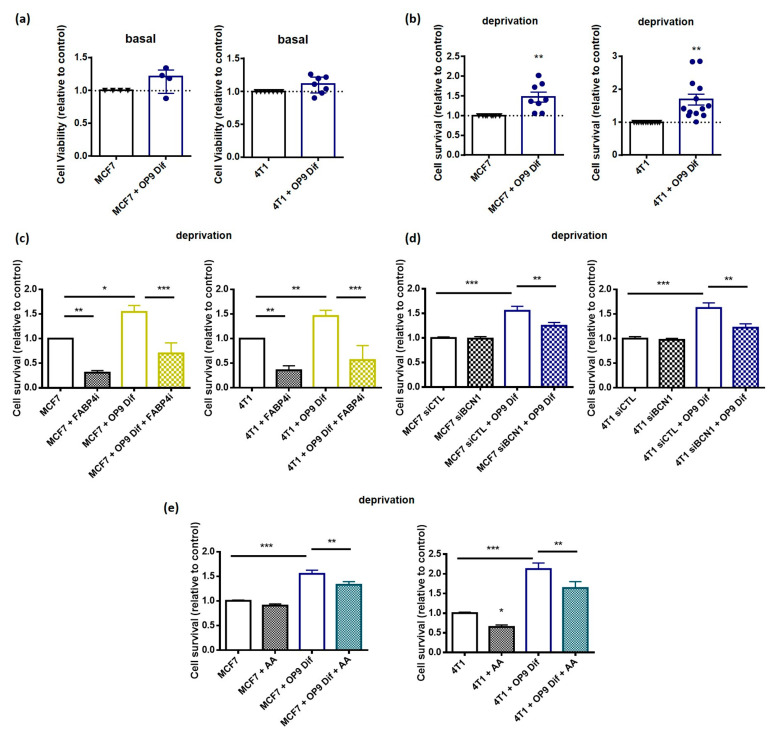
Activation of autophagy by adipocytes promotes cancer cell survival in nutrient-deprived conditions. (**a**) Cell viability assay in MCF7 and 4T1 cells with and without OP9 Dif in basal medium for 48 h (Median ± interquartile range, Wilcoxon Signed Rank Test ns *n* = 4–7). (**b**) Cell survival assay in MCF7 and 4T1 cells with and without OP9 Dif in HBSS deprivation medium for 48 h (Mean ± SEM, One sample *t* test ** *p* < 0.01, *n* = 8–13). (**c**) MCF7 and 4T1 cells with and without OP9 Dif in HBSS deprivation medium were treated by FABP4i (10 µM) for 48 h and cell survival assay was performed (Mean ± SEM, One-way ANOVA followed by Holm–Sidak’s multiple comparisons test * *p* < 0.05; ** *p* < 0.01 and *** *p* < 0.001, *n* = 5–6). (**d**) Cell survival assay of MCF7 and 4T1 cells transfected with siCTL and siBCN1 with and without OP9 Dif in HBSS deprivation medium for 48 h (Mean ± SEM, One-way ANOVA followed by Holm–Sidak’s multiple comparisons test ** *p* < 0.01 and *** *p* < 0.001, *n* = 9–10). (**e**) Cell survival assay of MCF7 and 4T1 cells treated with arachidonic acid (AA) 20 µM with and without OP9 Dif in HBSS deprivation medium for 48 h (Mean ± SEM, One-way ANOVA followed by Holm–Sidak’s multiple comparisons test * *p* < 0.05; ** *p* < 0.01 and *** *p* < 0.001, *n* = 11–14).

**Figure 7 cancers-13-03917-f007:**
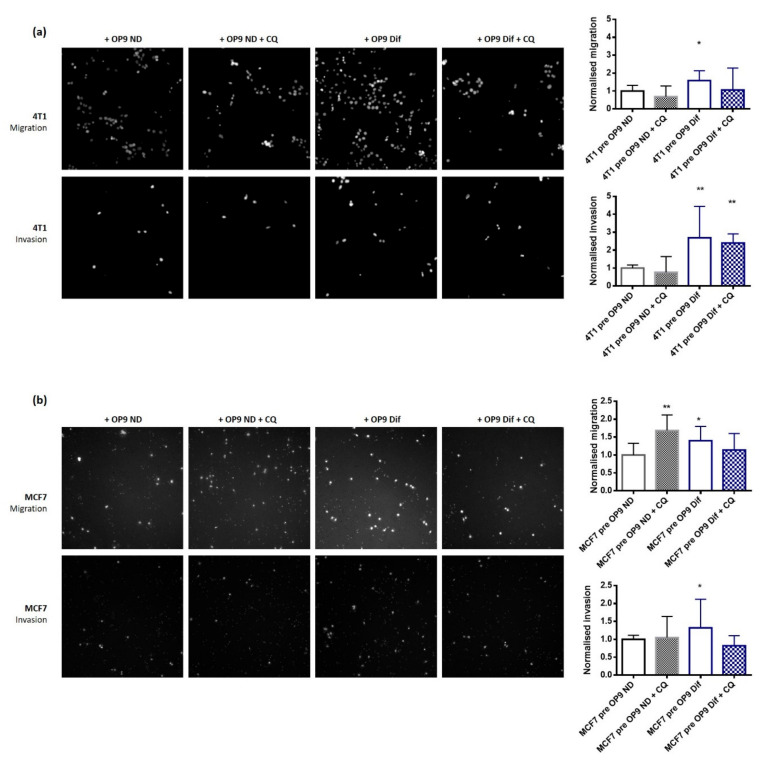
Activation of autophagy by adipocytes promotes cancer cell migration. (**a**,**b**) 4T1 (**a**) and MCF7 (**b**) cells were co-cultured with OP9 ND and OP9 Dif for 48 h and then subjected to cell migration and invasion assays with and without chloroquine (CQ) 20 µM for 24 h (Median ± interquartile range, Wilcoxon Signed Rank Test * *p* < 0.05 and ** *p* < 0.01, *n* = 12–14).

**Figure 8 cancers-13-03917-f008:**
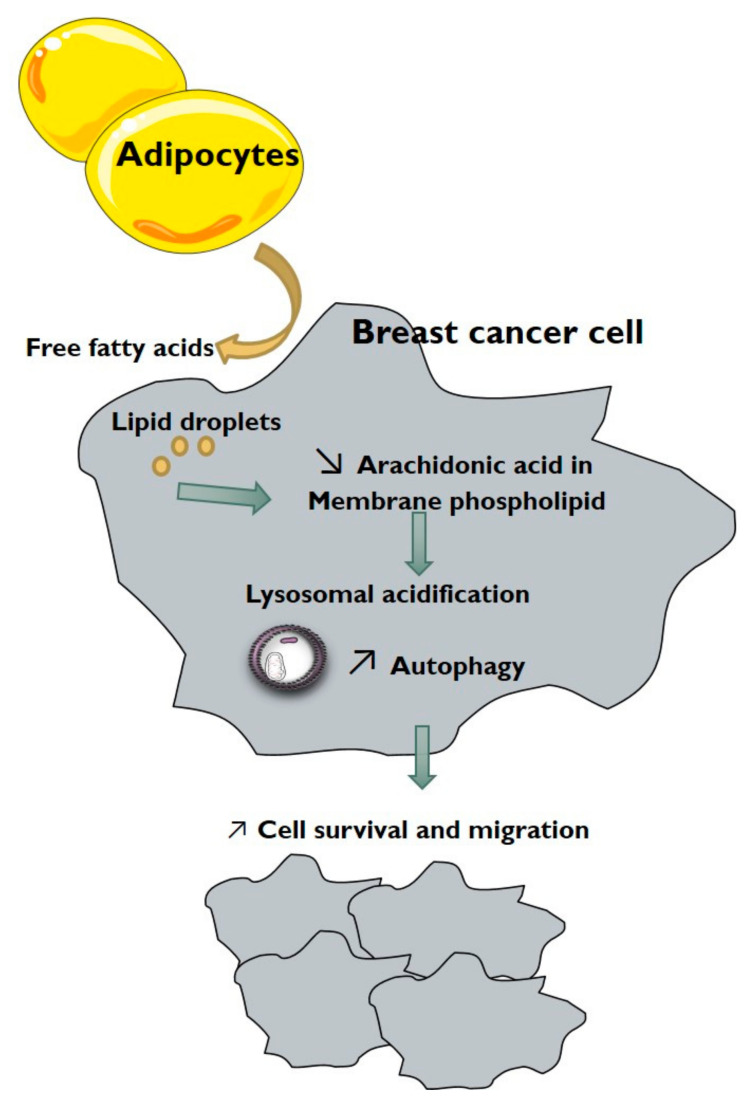
Schematic diagram depicting the interaction between adipocytes and breast cancer cells. The model describes the role of fatty acids in the modulation of autophagy in breast cancer cells and consequently in cancer cell survival and migration.

**Table 1 cancers-13-03917-t001:** Adipocyte-conditioned medium regulates fatty acid composition in breast cancer cells. Fatty acid ester composition in phospholipids from MCF7 and 4T1 cells treated with conditioned media from OP9 Dif and control media for 24 h. Data are expressed as the median percentage of total integrated peaks with range. Two-way ANOVA followed by Bonferroni’s multiple comparisons test, * *p* < 0.05; ** *p* < 0.01 and *** *p* < 0.001. SFA: saturated fatty acid, MUFA: monounsaturated fatty acid, PUFA: polyunsaturated fatty acid.

		4t1 Control	4t1 Condi	MCF7 Control	MCF7 Condi
Saturated					
Myristic Acid	14:0	2.34[2.03–3.44]	2.5[2.17–3.05]	2.92[2.24–3.39]	3.15[2.62–3.46]
Pentadecanoic Acid	15:0	0.33[0.27–0.53]	0.23[0.18–0.32]	0.27[0.1–0.32]	0.23[0.17–0.27]
Palmitic Acid	16:0	18.49[15.61–21.84]	18.95[15.54–20.18]	18.17[17.45–20.57]	17.96[17.05–20.42]
Margaric Acid	17:0	0.42[0.3–0.54]	0.28[0.14–0.45]	0.25[0.18–0.4]	0.2[0.11–0.37]
Stearic Acid	18:0	12.27[11.31–12.91]	12.35[11.24–12.81]	8.88[8.31–11.15]	8.69[7.81–9.98]
Arachidic Acid	20:0	0.25[0.14–0.5]	0.27[0.19–0.48]	0.22[0.11–0.31]	0.22[0.19–0.26]
Behenic Acid	22:0	0.52[0.23–0.64]	0.55[0.38–1.58]	0.41[0.1–0.78]	0.38[0.14–0.5]
Tricosanoic Acid	23:0	0.28[0.13–0.47]	0.2[0.02–0.41]	0.09[0.04–0.2]	0.1[0.02–0.22]
Lignoceric Acid	24:0	0.43[0.09–0.8]	0.38[0.2–0.81]	0.57[0.2–0.85]	0.31[0.24–0.52]
	Total SFA	35.27[39.84–31.85]	35.85[38.34–31.74]	32.11[35.82–30.65]	31.11[34.48–30.22]
Monounsaturated					
Myristoleic Acid	14:1	0.03[0.01–0.04]	0.03[0.02–0.06]	0.06[0.04–0.25]	0.09[0.04–0.2]
Pentadecenoic Acid	15:1	0.3[0.06–0.51]	0.21[0.05–0.29]	0.18[0.02–0.26]	0.3[0.02–0.4]
cis-7-Hexadecenoic Acid	16:1n-9	2.43[2.14–3.05]	2.65[2.39–3.38]	1.06[0.91–5.48]	0.99[0.85–5.86]
Palmitoleic Acid	16:1n-7	4.12[2.8–5.19]	5.12[4.37–7.15]	12.65[7.33–15.16]	14.04[6.99–14.83]
Heptadecenoic Acid	17:1	0.41[0.08–0.53]	0.29[0.07–0.4]	0.5[0.42–0.61]	0.48[0.34–0.52]
Oleic Acid	18:1	26.57[24.78–28.44]	23.89[20.43–31.5]	36.2[28.64–38.86]	37.62[35.63–39.86]
Gondoic Acid	20:1	0.89[0.54–1.26]	0.88[0.56–1.43]	0.4[0.31–0.7]	0.62[0.53–0.78]
Erucic Acid	22:1	0.29[0.13–0.45]	0.3[0.13–0.84]	0.26[0.12–0.41]	0.19[0.12–0.5]
Nervonic Acid	24:1	0.92[0.63–1.01]	0.64[0.16–1.18]	0.12[0.41–0.55]	0.12[0.5–0.57]
	Total MUFA	33.19[38.23–30.81]	31.15[42.48–28.11]	51.45[56.95–43.21]	55.26[56.59–52.06] ***
Polyunsaturated					
Linoleic Acid	18:2n-6	1.53[1.17–1.86]	1.09[0.88–1.51] **	1.51[1.1–1.8]	1.11[0.89–2.03]
Gamma Linolenic Acid	18:3n-6	0.1[0.03–0.2]	0.08[0.02–0.13]	0.04[0.02–0.07]	0.06[0.04–0.11]
Eicosadienoic Acid	20:2n-6	2.28[1.72–3.33]	2.24[1.77–3.26]	0.82[0.31–1.2]	0.7[0.34–1.16]
Dihomo-γ-linolenic Acid	20:3n-6	0.96[0.91–1.17]	0.81[0.67–0.91]	0.23[0.09–0.26]	0.16[0.06–0.25]
Arachidonic Acid	20:4n-6	3.8[3.24–4.09]	2.93[2.62–3.38] ***	2.04[1.41–2.66]	1.63[1.28–2.23] **
Docosadienoic Acid	22:2n-6	0.11[0.02–0.32]	0.14[0.02–0.18]	0.11[0.01–0.38]	0.09[0.04–0.21]
Docosatetraenoic Acid	22:4n-6	0.17[0.08–0.51]	0.08[0.05–0.4]	0.08[0.03–0.23]	0.06[0.03–0.11]
	Total n-6 PUFA	9.12[10.11–8.19]	7.64[8.46–6.42]	4.67[5.74–4.01]	4.25[4.76–3.41]
Alpha linolenic Acid	18:3n-3	0.09[0.05–0.28]	0.06[0.05–0.17]	0.05[0.03–0.11]	0.06[0.02–0.12]
Eicosatrienoic Acid	20:3n-3	0.1[0.03–0.13]	0.09[0.04–0.15]	0.04[0.01–0.09]	0.03[0.02–0.06]
Eicosapentaenoic Acid	20:5n-3	1.08[0.61–1.97]	0.99[0.65–1.6]	0.36[0.21–0.77]	0.23[0.11–0.4]
Docosapentaenoic Acid	22:5n-3	3.38[1.89–3.79]	2.24[1.73–3.18] ***	0.8[0.27–1.25]	0.53[0.27–0.93] **
Docosahexaenoic Acid	22:6n-3	3.11[2.1–3.98]	2.41[1.89–2.97] ***	1.64[1.18–2.23]	1.38[1.08–1.89] *
	Total n-3 PUFA	7.77[10.13–4.68]	5.91[7.69–4.98]	3.01[4.07–2.12]	2.18[3.16–1.53]

## Data Availability

This study includes no data deposited in external repositories. The authors confirm that the data supporting the findings of this study are available within the article and its Appendix A.

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
