# Peer review of "Adipocytes Promote Breast Cancer Cell Survival and Migration through Autophagy Activation"

_cancers, 2021, doi:10.3390/cancers13153917_

Round 1
Reviewer 1 Report
The authors claimed that adipocytes could increase autophagy in breast cancer cells and promote cell survival and cell migration.
Some comments:
- The three cell lines have different media, which media did the authors use for co-culture? Also, what is the effect of the selected medium on the growth of the other two cell lines?
- Is it reasonable for the authors to choose OP9 (murine-derived cells) and MCF7 (human-derived cells) for co-culture?
- The qPCR assay results were in the supplementary materials. The primers used table is not necessary to show in the text.
- In Figure 1, why did the authors choose the parameter “interquartile range” instead of SEM? And, the sample size is much lower than the others in this manuscript.
- The authors should show the results of the control group without OP9 cells.
- In the WB results, the LC3 have two proteins with different molecular weights. They are all LC3, or one is LC3I, and the other is LC3II?
- In figure 2, 4, and 6, the control group had too perfect replicate results and the authors need to provide more explanation.
- In figure 2, the authors stated that the number of samples is n = 13-14, however, the p62 results have more than 14 data points.
- In table 1, the first column and the comma in data were confusing.
Author Response
Reviewer 1
The authors claimed that adipocytes could increase autophagy in breast cancer cells and promote cell survival and cell migration.
We sincerely thank the reviewer for his comments and important suggestions. We now have made the requested changes, which, we think, have extensively improved the quality of the manuscript.
Some comments:
1 The three cell lines have different media, which media did the authors use for co-culture? Also, what is the effect of the selected medium on the growth of the other two cell lines?
For co-culture experiments, we maintained all cells in respective complete medium in order to avoid an effect due to a change in culture medium. Therefore, cell growth was similar. For clarification, we have added “respective” in the materials and methods description for the co-culture.
2 Is it reasonable for the authors to choose OP9 (murine-derived cells) and MCF7 (human-derived cells) for co-culture?
We agree with the reviewer that murine-derived cells might be questionable for human cancer model. However, to our knowledge no human cell line easily able to differentiate in adipocytes is available. For this reason, we have chosen OP9, which are murine-derived cells, to differentiate in adipocytes and co-culture with murine and human cancer cells. Furthermore, we have used two breast cancer cell lines; one murine to have a murine-murine model and one human. According to the reviewer 2 we have now added another human breast cancer cell line the MDA-MB-231. Therefore, we believe that the mechanism demonstrated in this article is similar between murine and human breast cancer cells.
3 The qPCR assay results were in the supplementary materials. The primers used table is not necessary to show in the text.
The table presenting the primers used for qPCR assay has been moved in supplementary materials.
4 In Figure 1, why did the authors choose the parameter “interquartile range” instead of SEM? And, the sample size is much lower than the others in this manuscript.
Data were represented according to the statistical test used (parametric or non-parametric) and the normality of the samples. We have rewritten the statistical part to clarify this point: “Statistical analyses were performed using GraphPad Prism V6 software. Normality was tested using D'Agostino & Pearson omnibus test. When normality test passed parametric tests were used: One sample t test or One-way ANOVA followed by Holm-Sidak's multiple comparisons test and data were expressed as mean +/- SEM. When normality test did not pass or was not possible for small sample experiments, non-parametric tests were used : Mann Whitney test, Wilcoxon Signed Rank Test or Kruskal-Wallis followed by Dunn's Multiple Comparison Test and data were expressed as median +/- interquartile range. Two-way ANOVA followed by Bonferroni's multiple comparisons test was used for multiple comparisons of fatty acid composition. Statistical tests and data representation are indicated in the figure legends. Statistical significance is indicated as: * p < 0.05; ** p < 0.01 and *** p < 0.001. NS stands for not statistically different.”
We have added two additional experiments for Figure 1C.
5 The authors should show the results of the control group without OP9 cells.
We have now compared co-culture with OP9 ND and without co-culture conditions. The co-culture of breast cancer cells with OP9 ND has no influence on autophagic flux (Supplemental Figure 2B,C), on autolysosome number (Supplemental Figure 2D), on intracellular degradation (Supplemental Figure 2E) and on lysosomal pH (Figure 3D).
6 In the WB results, the LC3 have two proteins with different molecular weights. They are all LC3, or one is LC3I, and the other is LC3II?
LC3 protein exists in two forms LC3I, which is cytosolic, and LC3II (lower molecular weight) which is LC3 conjugated to PE and bound to autophagosome. We have now indicated LC3I and LC3II in western blot representative images.
7 In figure 2, 4, and 6, the control group had too perfect replicate results and the authors need to provide more explanation.
In these figures, the data where normalised to the control with co-culture with OP9 ND for each independent experiment. Therefore, in control condition all values were 1. We have modified the graphic and figure legends indicating that they are normalized values.
8 In figure 2, the authors stated that the number of samples is n = 13-14, however, the p62 results have more than 14 data points.
We are sorry for this mistake; we were referring only to autophagic flux. We have corrected in figure legend n=13-14 for autophagic flux and n=19-20 for p62.
9 In table 1, the first column and the comma in data were confusing.
We have changed the comma in table 1 and added a column with the common name of the fatty acids in addition to the number of carbon and unsaturation.
Reviewer 2 Report
In this study, the authors proposed an interesting pathway of breast cancer cell survival and migration promotion through autophagy activation. Furthermore, the authors showed that mentioned pathway is promoted through adipocytes-cancer cells interaction. The studies mostly support this proposal, but some methodological and technical issues need clarification before this manuscript could be recommended for publication.
Major comments:
1). My biggest doubts are about the results that showed the role of adipocytes in cancer cell survival and migration. In fact, the authors based the most important thesis in the title based on research on one cell line (4T1). So why did the authors decide not to show the results for MCF-7 (I know from experience that these cells also show an invasive potential)? Moreover, to prove the thesis, the authors should show the results on a larger number of cell lines belonging to one species (at least 2-3 different human breast cancer cell lines).
2). Next, I am confused about the way the results are plotted on the charts. I do not understand why the authors decided to show the vast majority of the results normalized to control? In most cases, it is not essential, and I recommend showing absolute results (e.g. Fig6F, instead of the normalized value, the number of migrating cells should be shown). Also, I don't understand why the number of repetitions varies so much between individual experiences of the same type (e.g. Fig6A versus B, n=4-7 versus n=8-13).
3). Generally, why did the authors choose to use so many different kinds of statistical tests and especially in the same experiments (e.g. Fig.6A versus B, Wilcoxon Signed Rank Test versus One sample t-test)? I would like to ask for a justification for such a choice of statistical tests.
4). In Figure 6C and E, and in the 3.5 section. The authors stated that cell survival is prevented by AA or by an on-toxic concentration of FABP4 inhibitor supplementation of cancer cells co-cultured with OP9 Dif. These considerations should also take into account the fact that both factors alone inhibit proliferation in control cells. Therefore, authors should consider it both in the description of the results and in the discussion section. For example, after comparing the decrease in proliferation in MCF-7/MCF-7+FABP4i to MCF-7+OP9Dif/ MCF-7+OP9Dif+ FABP4i, it may turn out that the differences are not statistically significant at all.
5). More detailed information on the methodology should be given. E.g., I couldn’t find any information on how migrated cells were counted; how many cells were analyzed within 1 image per condition? How were specific housekeeping genes selected for RT-qPCR analysis (with a program like GeNorm, Normfinder or otherwise)? Also, please provide representative images for migration and invasion assays.
Minor comments:
1).I don't understand why the authors use the term “cell survival” to refer to normal cell proliferation. It seems to be confusing.
2). In line 491: The authors mentioned Supplementary Figure 5A, 5B, which are missed.
Author Response
Reviewer 2
In this study, the authors proposed an interesting pathway of breast cancer cell survival and migration promotion through autophagy activation. Furthermore, the authors showed that mentioned pathway is promoted through adipocytes-cancer cells interaction. The studies mostly support this proposal, but some methodological and technical issues need clarification before this manuscript could be recommended for publication.
We sincerely thank the reviewer for all the comments that definitely helped us to improve the manuscript. We now have made the requested changes, which, we think, have extensively improved the quality of the manuscript.
Major comments:
1). My biggest doubts are about the results that showed the role of adipocytes in cancer cell survival and migration. In fact, the authors based the most important thesis in the title based on research on one cell line (4T1). So why did the authors decide not to show the results for MCF-7 (I know from experience that these cells also show an invasive potential)? Moreover, to prove the thesis, the authors should show the results on a larger number of cell lines belonging to one species (at least 2-3 different human breast cancer cell lines).
We have previously decided to focus on 4T1 migration and invasion because they are known to be highly aggressive compared to MCF7 which have low migration capacity. We have now performed additional experiments with MCF7 on migration and invasion showing that the promotion of migration and invasion by adipocytes and the involvement of autophagy in migration is conserved (Figure 7A and 7B). In addition, we have confirmed in MDA-MB-231 that adipocytes promote autophagy (increase in autophagic flux and decrease in p62 in supplemental Figure 2A), migration and invasion (Supplemental Figure 6E).
2). Next, I am confused about the way the results are plotted on the charts. I do not understand why the authors decided to show the vast majority of the results normalized to control? In most cases, it is not essential, and I recommend showing absolute results (e.g. Fig6F, instead of the normalized value, the number of migrating cells should be shown). Also, I don't understand why the number of repetitions varies so much between individual experiences of the same type (e.g. Fig6A versus B, n=4-7 versus n=8-13).
We agree with the reviewer that showing absolute values is a better representation when possible. However, for most of biological experiments these absolute values can vary between independent experiments. This is especially the case for migration and invasion experiments, which are very sensitive experiments. In our case, the number of migrating cells in control condition varies between experiments mainly due to the batch of the inserts used and the passages of the cells. For this reason, we preferred to represent normalised values.
It is true that results presented in Figure 6A and 6B have different individual experiments. Since we were not able to see a difference in cell viability under basal conditions, we have decided to investigate more survival in deprivation conditions. Because we focused on cell survival in different conditions, we had several additional experiments that we were able to include in the graph.
3). Generally, why did the authors choose to use so many different kinds of statistical tests and especially in the same experiments (e.g. Fig.6A versus B, Wilcoxon Signed Rank Test versus One sample t-test)? I would like to ask for a justification for such a choice of statistical tests.
For each set of data and if we had more than 5 values we have first tested whether the distribution was Gaussian and then use the appropriate parametric or non parametric tests. For Figure 6A since the number of values was below 5 we have used non parametric tests whereas with more values in Figure 6B we were allowed to use parametric tests. We have clarified this aspect in the statistical part of the materials and methods: “Statistical analyses were performed using GraphPad Prism V6 software. Normality was tested using D'Agostino & Pearson omnibus test. When normality test passed parametric tests were used: One sample t test or One-way ANOVA followed by Holm-Sidak's multiple comparisons test and data were expressed as mean +/- SEM. When normality test did not pass or was not possible for small sample experiments, non parametric tests were used: Mann Whitney test, Wilcoxon Signed Rank Test or Kruskal-Wallis followed by Dunn's Multiple Comparison Test and data were expressed as median +/- interquartile range. Two-way ANOVA followed by Bonferroni's multiple comparisons test were used for multiple comparisons of fatty acid composition. Statistical tests and data representation are indicated in the figure legends. Statistical significance is indicated as: * p < 0.05; ** p < 0.01 and *** p < 0.001. NS stands for not statistically different."
4). In Figure 6C and E, and in the 3.5 section. The authors stated that cell survival is prevented by AA or by an on-toxic concentration of FABP4 inhibitor supplementation of cancer cells co-cultured with OP9 Dif. These considerations should also take into account the fact that both factors alone inhibit proliferation in control cells. Therefore, authors should consider it both in the description of the results and in the discussion section. For example, after comparing the decrease in proliferation in MCF-7/MCF-7+FABP4i to MCF-7+OP9Dif/ MCF-7+OP9Dif+ FABP4i, it may turn out that the differences are not statistically significant at all.
We understand the reviewer point and indeed the effects of FABP4 inhibitor and AA alone are important. We have modified the result description and the discussion accordingly.
5). More detailed information on the methodology should be given. E.g., I couldn’t find any information on how migrated cells were counted; how many cells were analyzed within 1 image per condition? How were specific housekeeping genes selected for RT-qPCR analysis (with a program like GeNorm, Normfinder or otherwise)? Also, please provide representative images for migration and invasion assays.
We have added more information on the methodology:
For migration experiment, “The total number of nuclei per image was counted after thresholding using the particle analysis tool of ImageJ.”
For intensity analysis, the total intensity was measured and normalised by the number of cells on that field. This has been performed on minimum 5 images per independent experiment. “Fluorescence intensity was measured using ImageJ and normalized to cell number observed using bright field on minimum 5 images per condition”
For counting of autophagosomes and autolysosomes “Yellow and red vesicles were counted using the Red and Green Puncta Colocalization Macro for ImageJ (developed by Daniel J. Shiwarski, Ruben K. Dagda and Charleen T. Chu) to identify autophagosome and autolysosome number on minimum 5 images and 15 cells per condition.”
“HPRT and β-actin were used as housekeeper genes after selection using Normfinder.”
Representative images for migration and invasion assays are now provided in Figure 7A, 7B and Supplemental Figure 6E.
Minor comments:
1).I don't understand why the authors use the term “cell survival” to refer to normal cell proliferation. It seems to be confusing.
We agree that this could be confusing. We wanted to avoid the term “cell proliferation” because the SRB test we used is not a true cell proliferation assay but an evaluation of cell number. To clarify this point, we have changed assays using complete medium by “cell viability” and we kept “cell survival” for assays using deprivation. In this case, we evaluate the number of cells able to survive to deprivation condition therefore less cells remained compared with the beginning of the experiment.
2). In line 491: The authors mentioned Supplementary Figure 5A, 5B, which are missed.
We are sorry for that. We have now included this figure in the submission.
Round 2
Reviewer 1 Report
no comments
Reviewer 2 Report
I appreciate the authors put much effort to fix the issues that were mentioned in the original review. The corrections implemented by the Authors improve the quality of the manuscript, and the manuscript could be recommended for publication.